# Cholecystokinin Receptor Antagonist Induces Pancreatic Stellate Cell Plasticity Rendering the Tumor Microenvironment Less Oncogenic

**DOI:** 10.3390/cancers15102811

**Published:** 2023-05-18

**Authors:** Gurbani Jolly, Tetyana Duka, Narayan Shivapurkar, Wenqiang Chen, Sunil Bansal, Amrita Cheema, Jill P. Smith

**Affiliations:** 1Department of Oncology, College of Medicine, Georgetown University, Washington, DC 20007, USA; 2Department of Medicine, College of Medicine, Georgetown University, Washington, DC 20007, USA

**Keywords:** tumor microenvironment, fibrosis, pancreatic cancer, desmoplasia, CCK

## Abstract

**Simple Summary:**

The tumor microenvironment of pancreatic cancer consists of dense fibrotic stroma, which, to a certain extent, accounts for the relative drug resistance of this malignancy. Activated pancreatic stellate cells (PSC) and myofibroblasts are responsible for this fibrosis, and strategies to target the fibrosis or render the stellate cells quiescent have been investigated in order to improve therapies for pancreatic cancer. Many have studied TGF-β and other pathways involved in fibrosis. In this work, we study the role of a novel cholecystokinin receptor signaling pathway in pancreatic cancer fibrosis. Treatment with a CCK receptor antagonist, proglumide, induced plasticity of activated mouse and human fibroblasts to revert the cells to a quiescent state with decreased migration, proliferation, and production of collagenous proteins of the tumor microenvironment. Interruption of the CCK-B receptor pathway provides a novel strategy to alter the extracellular matrix in pancreatic cancer by changing PSCs from an activated state to quiescence.

**Abstract:**

CCK receptors are expressed on pancreatic cancer epithelial cells, and blockade with receptor antagonists decreases tumor growth. Activated pancreatic stellate cells or myofibroblasts have also been described to express CCK receptors, but the contribution of this novel pathway in fibrosis of the pancreatic cancer microenvironment has not been studied. We examined the effects of the nonselective CCK receptor antagonist proglumide on the activation, proliferation, collagen deposition, differential expression of genes, and migration in both murine and human PSCs. CCK receptor expression was examined using western blot analysis. Collagen production using activated PSCs was analyzed by mass spectroscopy and western blot. Migration of activated PSCs was prevented in vitro by proglumide and the CCK-B receptor antagonist, L365,260, but not by the CCK-A receptor antagonist L365,718. Proglumide effectively decreased the expression of extracellular matrix-associated genes and collagen-associated proteins in both mouse and human PSCs. Components of fibrosis, including hydroxyproline and proline levels, were significantly reduced in PSC treated with proglumide compared to controls. CCK peptide stimulated mouse and human PSC proliferation, and this effect was blocked by proglumide. These investigations demonstrate that targeting the CCK-B receptor signaling pathway with proglumide may alter the plasticity of PSC, rendering them more quiescent and leading to a decrease in fibrosis in the pancreatic cancer microenvironment.

## 1. Introduction

Pancreatic ductal adenocarcinoma (PDAC) will soon become the second leading cause of cancer-related deaths in the USA [1] and has the poorest prognosis of all solid tumors [2]. Reasons for the dismal prognosis include the lack of tests for early detection and the cancer’s relative resistance to chemotherapy and immunotherapy [3,4]. One reason why PDAC is ‘resistant’ to immune checkpoint antibodies and chemotherapeutic agents has been attributed to the dense fibrosis of the tumor microenvironment (TME) [5,6,7] that prevents permeation of many agents and penetration of effector T-killer lymphocytes [8,9]. Therefore, investigators have turned their attention to seeking strategies to remodel the TME to render this cancer more susceptible to therapy [10,11,12,13]. Many of these agents that targeted the PDAC fibrosis [14] or immune components, such as the CCL2-CCR2 axis or IL-10 [15,16], have unfortunately failed in the clinic. Understanding pancreatic cancer and its microenvironment is very complex, and these clinical investigations have shown us that interruption of one component of the TME alone is not adequate to inhibit the growth and metastases of this aggressive tumor. The various cells of the TME cross-talk through signaling mechanisms and metabolic pathways [17], and blockade of just one cell type seems to enhance immune escape mechanisms that make pancreatic cancer very difficult to treat [18].

Pancreatic stellate cells (PSCs) and activated myofibroblasts have been implicated for their role in creating the dense fibrosis of the tumor microenvironment [19,20]. Multiple studies have shown that PSCs can create a pro-tumorigenic environment through interaction with pancreatic cancer epithelial cells resulting in PSC proliferation, migration, extracellular matrix (ECM) remodeling, and immunosuppression, among other factors [21,22,23]. It is thought that PSCs are quiescent in the normal pancreas, with its distinctive presence of Vitamin A-containing droplets in the cytoplasm [24]. However, upon response to injury or inflammation, these quiescent PSCs can morphologically transform into the activated myofibroblast phenotype, characterized by expression of ECM components such as α-smooth muscle actin (αSMA), collagen, growth factors, cytokines, and chemokines which can affect the penetration of drug therapeutics [23,25]. ECM-modifying enzymes, such as matrix metalloproteinases (MMPs) that degrade ECM components to promote invasion have been associated with tumor progression and the aggressive behavior of pancreatic cancer [24,26].

In the pancreatic cancer microenvironment, these transformed fibroblasts or cancer-associated fibroblasts (CAFs) have a heterogeneous population [22], including inflammatory CAFs, myofibroblast CAFs, and even antigen-presenting CAFs [22,27]. CAFs have been demonstrated to play several roles, including promoting tumor growth and metastasis formation, depositing extracellular matrix (ECM), and establishing an immuno-suppressive microenvironment. Recent evidence indicates that CAF subtypes are dynamic and also exhibit plasticity with the ability to interconvert depending on prompts from tumor cells, culture conditions, and therapeutic regimens. Therapeutic strategies are being applied to eliminate fibrosis by targeting different signaling pathways in PSCs. Elimination of sonic hedgehog (SHH) signaling depletes the myofibroblasts, which predominately express αSMA, within the TME; however, this complete elimination rendered PDAC more aggressive and metastatic [28]. Hence, novel strategies are needed to safely revert activated PSC to the quiescent state without disruption of normal physiologic pathways.

Our research laboratory has found that the G-protein coupled rhodopsin receptor, the cholecystokinin-B receptor (CCK-BR), is upregulated in pancreatic cancer and plays an important role in cancer growth and fibrosis [29,30,31]. Additionally, CCK receptors have been reported on isolated rat and human pancreatic stellate cells and repeated CCK administration has been shown to activate PSCs [32,33]. Two classical CCK receptors have been previously identified, the CCK-A (CCK-1) and the CCK-B (CCK-2) receptors [34,35]. Selective CCK receptor antagonists have been developed [36] that inhibit signaling at either the CCK-A receptor or the CCK-B receptor. Proglumide is a unique nonselective CCK receptor antagonist that is water soluble and has an affinity for both CCK receptor types [37]. Treatment of mutant Kras mice or pancreatic cancer-bearing mice with proglumide has previously been shown to decrease fibrosis in the ECM [31] and the tumor microenvironment [38]. By decreasing pancreatic tumoral fibrosis, we demonstrated that proglumide improved the efficacy of chemotherapy [39] and immune checkpoint antibody therapy with an anti-PD1 antibody [40] in murine models of pancreatic cancer. Although murine models are often used to study pancreatic cancer and the TME, CCK receptors have not been characterized in murine PSCs. Furthermore, the mechanism by which proglumide reverses fibrosis in the pancreatic tumor microenvironment is unknown. We hypothesize that proglumide induces plasticity of the activated myofibroblasts in the pancreatic tumor microenvironment, transforming these cells to a quiescent state. The purpose of this investigation was to examine the role of the CCK receptor signaling pathway as a novel approach to induce PSC plasticity.

## 2. Materials and Methods

### 2.1. Cell Lines

Immortalized mouse pancreatic stellate cells (mPSCs) were a gift from Dr. Mathison (Mayo Clinic, College of Medicine, Rochester, MN, USA) [41]. These murine cells were isolated from C57BL/6 mice and immortalized by incubation with an ecotropic retrovirus containing SV40 large T-antigen. The mPSCs were cultured in complete DMEM (Gibco, Miami, FL, USA; Cat #12491), supplemented with 10% FBS, 1% L-Glutamine, and 1% pen/strep in humidified air with 5% CO_2_. Human pancreatic stellate cells (hPSC) were purchased from ScienCell Research Laboratories in (Carlsbad, CA, USA; Catalog #3830). These cells are negative for HIV-1, HBV, HCV, mycoplasma, bacteria, yeast, and fungi. These cells were grown in RPMI (Invitrogen, Grand Island, NY, USA) and medium supplemented with 10% fetal bovine serum (Invitrogen, Waltham, MA, USA). All cells were grown in vitro with 5% CO_2_ and humidified air.

### 2.2. Peptides and Receptor Antagonists

Cholecystokinin-10 (CCK, cerulein) was purchased from Sigma-Aldrich, Inc. (St. Louis, MO, USA; Cat #17650-98-5). The following CCK receptor antagonists were purchased from Tocris Bioscience (Bristol, UK): Proglumide (Cat #1478/50), the CCK-A receptor antagonist L364,718 (Cat #2304), and the CCK-B receptor antagonist L365,260 (Cat #2767). Peptides or antagonists were prepared in a stock solution and sterilized with a 0.2 µm filter for in vitro cultures.

### 2.3. Relative Gene Expression by Quantitative RT-PCR

Total RNA was extracted from cultured mPSC and hPSC to determine the expression of differentially expressed genes (DEGs) associated with the pancreas TME using the Qiagen miRNeasy kit (Qiagen, Germantown, MD, USA; Cat #217004). Synthesis of cDNA was performed using an RT first strand kit (Qiagen, Cat#330401). Quantitative Reverse Transcription PCR (qRT-PCR) was performed using RT SYBR Green ROX qPCR Mastermix (Qiagen, Cat# 330520) in a 7300 Real-Time PCR System (Applied Biosystems, Waltham, MA, USA) programmed at 1 cycle at 95 °C for 10 min followed by 40 cycles at 95 °C for 15 s and 60 °C for 1 min in the presence of mouse (Table 1) or human (Table 2) specific primers. *Hprt* served as the normalizer control for mouse samples and *GAPDH* for the human samples. Replicates of *n* = 3 were performed for each gene of interest. Single amplicons were confirmed by PCR dissociation curves.

### 2.4. Migration Assay

Twelve-well plates (Falcon, Glendale, AZ, USA; Cat #353043) were seeded with mPSCs or hPSCs and complete media mixture and incubated for approximately 48 h until 100% confluent. Once confluent, cells were divided into four treatment groups (*n* = 6 wells/ea): control (media alone), CCK (10 nM), proglumide (20 nM), and combination (10 nM CCK and 20 nM proglumide). A “mock wound” was created by scratching a 20 µL sterile tip from the 12 o’clock to 6 o’clock position. Mouse PSCs were treated for 26 h, with images being recorded at T = 0, 2.5, 3.5, 5, 6.5, and 26 h. The same protocol was repeated in the mPSCs (*n* = 6 wells) treated with proglumide (20 nM) and the selective CCK receptor antagonists: CCK-AR antagonist, L364,718 (1 nM) and CCK-BR antagonist, L365,260 (1 nM). Receptor antagonist-treated cells were treated for 26 h, with images being recorded at T = 2.5, 6.5, and 26 h.

Human PSCs were also plated and grown to confluency, and the migration assay was performed as above; however, the hPSCs grow more slowly, so the width of the scratch was recorded after 24, 48, and 72 h after treatment with CCK (10 nM), proglumide (20 nM), or the combination therapy. PSCs were imaged at 4× objective magnification with a Moticam 1sp: 1.3mp digital camera (Kowloon, Hong Kong).

Images captured were saved as JPEG files and analyzed using ImageJ version 1.46 to determine the scratch length at each time interval. A total of 6 measurements were taken per image to determine the average pixel length of the scratch. Statistical analyses were performed using GraphPad Prism software version 9.0 (San Diego, CA, USA). Comparisons between scratch lengths were calculated using unpaired, parametric Student’s *t*-tests, and Bonferroni correction was applied for multiple comparisons to control.

### 2.5. Western Immunoblotting

CCK-B receptors, CCK-A receptors, and downstream signaling activation were also confirmed in mPSC by western analysis. The mPSC were grown to log phase then treated with media alone (control), CCK (10 nM), proglumide (20 nM), L364,718 (1 nM), and L365,260 (1 nM) for 24 h. Protein lysates were collected and analyzed with gel electrophoresis (as described in detail below) and probed with a rabbit polyclonal CCK-BR antibody (Invitrogen, Cat # PA5-77377) or a goat polyclonal CCK-AR antibody (Invitrogen, Cat# PA5-18385) at a titer of 1:200 overnight at 4 °C. Blots were stripped (as below) and incubated with anti-mouse rabbit polyclonal AKT antibody (Invitrogen, Cat # 44-609G) at 1:1000 titer and anti-mouse rabbit monoclonal Phospho-AKT1 (Ser473) Monoclonal Antibody (clone, 14-6) (Invitrogen, Cat # 44-621G) at a titer of 1:1000. Blots were incubated at room temperature for 60 min with a rabbit HRP-linked secondary antibody (titer 1:5000, Invitrogen, Cat #31460).

Protein lysates were collected from control, CCK (10 nM), proglumide (20 nM), combination, and selective CCK receptor antagonists (1 nM) treated mPSCs. Cells were homogenized with a RIPA buffer (Pierce, Waltham, MA, USA; Cat #89900) and protease inhibitors (Pierce, Cat #A32955). Protein concentration was compared using a BSA Protein Assay Kit (Pierce, Cat #23225) standard curve solutions at 0, 25, 50, 125, 250, 500, and 1000 µg/mL with dye reagent concentrate. Samples were incubated at 37 °C for 30 min and read at 490 nm on the VMax Kinetic ELISA Absorbance Microplate Reader (Agilent BioTek, Santa Clara, CA, USA). A linear regression curve fit was calculated (R^2^ = 0.9911) using the standard curve solutions and samples, yielding a protein concentration of 8–48 µg among mPSC samples.

A multi-colored broad protein range ladder (20 µL) (Spectra Biolabs, San Diego, CA, USA; Cat #26634) and protein (25 µg) were loaded to wells of NuPAGE 4–12% Bis-Tris gels (Invitrogen, Cat #NP0321BOX). Proteins were separated by electrophoresis with 1× Running buffer—1:19 20× MOPS SDS Running Buffer (NuPage, Cat #NP001) and deionized water, respectively,—at a constant 150 V for 50 min. Gels were transferred onto nitrocellulose membranes (ThermoScientific, Waltham, MA, USA; Cat #88018) with 1× transfer buffer—1:2:17 20× Transfer Buffer (NuPage, Cat #NP0006-1), methanol, and deionized water, respectively, at a constant 15 V overnight at 4 °C. Blots were blocked for 60 min with 5% milk in 1× TBST, then incubated overnight at 4 °C with primary antibody αSMA (1:500) (Abcam, Cambridge, UK; Cat #ab5694) in 5% milk. Blots were incubated at room temperature for 60 min with a rabbit HRP-linked secondary antibody (1:5000, Invitrogen, Cat #31460), sprayed with WesternBright ECL Spray (VWR, Radnor, PA, USA; Cat #K-12049-D50), and read on the Amersham Image 600 (Temecula, CA, USA). Membranes were stripped with stripping buffer (ThermoScientific; Cat #46430) and incubated with 5% milk for 60 min at 4 °C. The same incubation, exposure, and striping processes were repeated with Col1α1 (1:500) (Invitrogen; Cat #PA5-89281), β-actin antibody (1:500) (Invitrogen; Cat #MA1-140), and β-tubulin antibody (1:500) (Invitrogen; Cat #PA5-21826). Bands for αSMA, Col1α1, and β-actin were quantified using ImageJ. Since the blots had been stripped three times previously, β-tubulin expression was not utilized for comparison calculations. Statistical analyses were performed using GraphPad Prism software version 9.0. Comparisons between integrated densities were calculated using unpaired, parametric *t*-tests.

### 2.6. Proline and 4-Hydroxyproline Analysis by Mass Spectroscopy

The solvents acetonitrile, water, isopropanol, and methanol were purchased from Optima grade (Fisher Scientific, Hampton, NH, USA); high purity formic acid (ThermoScientific); proline, 4-hydroxyproline, and proline-2,5,5-d3 (Sigma-Aldrich). The mPSCs cell samples were vortexed with PBS (25 µL) and heat-shocked three times for 30 s at −80 °C followed by 90 s at 37 °C. An extraction buffer (200 µL) with internal standard (1 µg/mL of proline-2, 5, 5-d3) was added to each sample and vortexed, followed by sonication. Samples were incubated on ice for 20 min, −20 °C for 20 min, and centrifuged at 13,000 rpm for 20 min at 4 °C. Supernatants were transferred to the sample vial for Ultra-Performance Liquid Chromatography Mass Spectrometry (UPLC-MS) analysis. A pooled QC sample was generated by mixing 30 µL of each sample.

Untreated and 24-h proglumide (20 nM) treated mPSCs (*n* = 6 each) were analyzed for proline and 4-hydroxyproline metabolite levels by mass spectroscopy in the Georgetown University Lombardi Core Metabolomics and Proteomics facility. An MRM-based mass spectrometry method was developed for the measurement of proline and 4-hydroxyproline by UPLC-MS system. The samples were resolved on an Acquity UPLC BEH C18, 1.7 µm, 2.1 × 100 mm column online with a triple quadrupole mass spectrometer (Xevo-TQ-S, Waters Corporation, Milford, MA, USA) as described previously [42]. Signal intensities from the MRM Q1 > Q3 ion pairs for the drug proline (116.1 > 69.9), 4-hydroxyproline (132.1 > 67.9), and proline-2,5,5-d3 (119 > 70) were ranked to ensure selection of the most intense precursor and fragment ion pair for MRM-based quantitation. The lower limit of detection (LOD) for proline and 4-hydroxyproline was found to be 0.5 ng/mL and 0.25 ng/mL, respectively. The linear dynamic range for quantification was determined at 1.0 µg/mL to 1 ng/mL and 5.0 µg/mL to 0.5 ng/mL for proline and 4-hydroxyproline, respectively. The mass spectrometry parameters and data acquisition details have been included in the Appendix A. Appendix A shows a sample chromatogram of eluted proline, 4-hydroxyproline, and internal standard (proline-2,5,5-d3) performed by UPLC-MRM-based mass spectrometric analysis. A calibration curve for proline is provided in Appendix A, and the corresponding data are in Appendix A. A calibration curve for hydroxyproline is provided in Appendix A, and the corresponding data are in Appendix A.

### 2.7. Stellate Cell Proliferation Assay

Cell proliferation experiments were performed using the 3-(4,5-methylthiazol-2-yl) −2,5-diphenyltetrazolium bromide (MTT) assay. In the MTT assay, the mouse or human pancreatic stellate cells were seeded with RPMI that contained 10% FBS at a density of 0.1 × 10^6^/well in 12-well plates, grown overnight, washed in PBS, and incubated with serum-free medium for 24 h. Cells were treated with CCK (10 nM), proglumide (20 nM), and the combination (CCK and proglumide). After 24 h, MTT was added (0.5 mg/mL, a final concentration per well) for 4 h. Formazan products were solubilized with DMSO, and the optical density was measured at 570 nm. The data from the proliferation assays were expressed as a percentage of the control counts, which were set as having 100% viability. The proliferation of the hPSCs was also repeated, and cells were allowed to grow for 24 and 48 h since these cells grow more slowly than the mPSC.

### 2.8. Statistics

Migration assays were performed with replicates of 3, and a minimum of *n* = 6 measurements of scratch width were recorded using ImageJ software version 1.46, and mean pixel diameters were calculated between groups with two-way analysis of variance using GraphPad (Prism) software program version 9.0. Quantitative RT-PCR expression analyses were calculated using Minitab version 19 with a Student’s *t*-test on the normalized mean ΔCT (the difference between the cycle counts of the gene of interest minus the count of an endogenous control) values for each group, with Bonferroni corrections applied to adjust for multiple comparisons. Western blot quantification of protein expressions was analyzed by densitometry, and each sample was normalized by corresponding β-actin or β-tubulin performed on ImageJ. MTT proliferation assays were performed in duplicate with *n* = 12 wells per treatment group. Treatment groups were compared using a two-way ANOVA test with Prism version 9.0. Mass Spectroscopy calibration curves are described above. Analysis of pair-treated or untreated mPSCs was undertaken using a Student’s *t*-test with significance set at 95% confidence or *p* < 0.05.

## 3. Results

### 3.1. PSCs Express CCK-A and CCK-B Receptor Protein by Western Blot

Phillips et al. [33] have previously shown mRNA expression or CCK-AR and CCK-BR in hPSC, but investigators have not previously described CCK receptors on mouse PSCs. Herein, we confirmed the protein expression for CCK-AR and CCK-BR in control or untreated mPSCs (Figure 1A). When cultured mPSCs were treated with the agonist CCK or the CCK-BR antagonist, there was increased expression of CCK-BR expression in the mPSCs (Figure 1B), hence, confirming the normal physiologic reaction. We have previously shown that the CCK-BR expression becomes upregulated in mice treated with a high-fat diet that raised endogenous CCK blood levels [43]. When mPSCs were treated with CCK peptide, the CCK-AR expression increased yet did not reach statistical significance. When mPSCs were treated with the CCK-AR antagonist, the CCK-AR protein expression was significantly downregulated (Figure 1C), suggesting possibly that the function of the CCK-AR is different in the PSC than the CCK-BR since the expression is different after exposure. Treatment of mPSC with CCK activates downstream AKT phosphorylation (Figure 1D). The CCK-BR selective antagonist also slightly increased phosphorylation of downstream AKT in mPSC, but this value did not quite reach statistical significance.

### 3.2. Proglumide Therapy Significantly Reduced PSC Migration

The migration rate of mPSCs over time is shown in Figure 2A. After 6 h, CCK peptide treatment significantly increased the migration rate of mPSC compared to control cells (*p* = 0.01). By 26 h, untreated control and CCK-treated cells migrated to completely close the gap, but the gap remained open in the cells treated with proglumide therapy (*p* ≤ 0.01) alone or in combination with CCK implying that proglumide rendered the mPSC more quiescent and slowed migration. The mean baseline scratch width was not different between the groups (Figure 2B), and final mean differences were observed at 26 h. Representative images from the migration assay are shown in Figure 2C for each of the treatment groups and untreated control mPSCs at baseline and after 26 h.

Migration of human PSCs was also affected by treatment with CCK and proglumide. Although these primary cell cultures migrate more slowly than the immortalized mPSC, CCK significantly accelerated the migration at 24 and 48 h (Figure 2D; *p* < 0.0001), and proglumide therapy significantly slowed the migration of hPSCs compared to untreated controls (Figure 2D; *p* < 0.0001). By 72 h, the gaps had closed with all the treatments. Representative images of each treatment group over time are provided in Appendix A.

Since proglumide has binding affinity for both the CCK-AR and the CCK-BR, in order to determine whether proglumide’s inhibitory effects on PSC migration were mediated predominantly by the CCK-AR or the CCK-BR, the mPSC migration assay was repeated with selective CCK receptor antagonists. We found that only proglumide and L365,260, the CCK-BR antagonist, blocked mPSC migration, whereas the CCK-AR antagonist, L364,718 did not (Figure 2E). Although the CCK-AR antagonist-treated cells migrated slightly slower than the untreated control PSCs and did not completely close the gap over 26 h, the difference compared to untreated PSCs did not reach significance (*p* = 0.08). These data support that the effects proglumide has on the migration of PSCs are primarily mediated by the CCK-BR. Representative images of each treatment group for the antagonist migration experiment over time are provided in Appendix A.

### 3.3. Effects of CCK and Proglumide Treatment on mPSCs and hPSCs Alters Differently Expressed Genes Associated with the Pancreatic Extracellular Matrix (ECM)

Many genes associated with hepatic stellate cell activation in the liver and the liver ECM have been described, but selective genes associated with PSCs and the pancreas ECM have not been extensively studied. For example, *Ephb2*, Ephrin type-B receptor 2, has been shown to regulate hepatic fibrosis in hepatic stellate cells, among other processes such as proliferation, migration, and angiogenesis [44]. In this study, *Ephb2* downregulated CCK-activated expression in mPSCs (*p* < 0.05; Figure 3A). m*Rictor* (rapamycin-insensitive companion of mammalian target of rapamycin) and m*Rheb* (Ras homolog enriched in brain) have been associated with the mTORC pathway regulation and may play a role in fibrosis through hepatic stellate cell activation [45]. One study noted that reduction of both m*Rictor* and m*Rheb* through RNAi methods showed a significant decrease in *Acta2* (which codes for α-smooth muscle actin) and *Col1α1* (which codes for collagen type 1α chain 1). We found that proglumide treatment of mPSCs decreased CCK-activated expression of m*Rictor* (*p* = 0.006) in mPSCs compared to CCK therapy (Figure 3B). Additionally, pro-inflammatory cytokine IL-8 is produced by pancreatic stellate cells and is often associated with invasive tumors and fibrosis [46]. Upon treatment with combination therapy, m*IL*-*8* expression was significantly decreased in mPSCs as compared to control (*p* = 0.042) and CCK therapy (*p* = 0.008; Figure 3C). Gli-3, a Glioblastoma family protein, is involved in Hedgehog (Hh) signaling and can act as a transcriptional activator of multiple pathways involved in invasion and epithelial-mesenchymal transition [47]. *Gli*-*3* mRNA downregulation has been associated with decreases in profibrotic markers such as vimentin and connective tissue growth factor [48]. Treatment with proglumide significantly downregulated CCK-stimulated m*Gli*-*3* expression in mPSCs by qRT-PCR (*p* < 0.05) compared to both control and CCK treatment (Figure 3D). Activated pancreatic stellate cells have classically been identified through changes in the ECM, particularly an increased expression of αSMA (coded by *Acta2*) and collagen type 1 (coded by Col1α1), which has been associated with fibrosis and poor PDAC prognosis [49]. Based on qRT-PCR, proglumide therapy decreased *Acta2* expression, but this decreased and did not reach statistical significance (Figure 3E). However, *Col1α1* expression (Figure 3F), a key gene involved in mPSC fibrosis, was significantly decreased. These results support that proglumide may decrease pancreatic fibrosis by downregulation of fibrosis-associated genes.

Other fibrosis-associated gene mRNA expression did not significantly change with proglumide therapy in mPSC, including *Hic*-*5* (hydrogen peroxide-inducible clone 5)—associated with hepatic fibrosis and stellate cell activation via TGF-β [50]; *Fap* (fibroblast activating protein gene)—not usually expressed in adult tissues but associated with fibrosis, tissue remodeling, and pancreatic cancer [51]; and *IL*-*1β*, a pro-inflammatory cytokine mRNA, associated with PSC activation and fibrosis via TGF-β [52] (Appendix A). Collagen type IV, involved in the basement membrane structure, has been shown to be elevated in pancreatitis, with the potential to be used as a predictive biomarker [53]. *Col4α* mRNA expression was significantly decreased with proglumide (*p* = 0.029) treatment (Appendix A).

Human PSCs showed similar patterns of differentially expressed genes associated with the ECM when treated in culture with CCK, proglumide, or the combination of both CCK and proglumide as in the mPSC. In the hPSC, proglumide therapy significantly decreased the expression of all the genes associated with the ECM, whether given as monotherapy or in combination with CCK. In comparison to the mouse PSCs, the downregulation of DEGs associated with ECM in human PSCs, was slightly more pronounced, and proglumide also decreased any stimulatory effects of CCK (Figure 4A–F). In hPSC, *ACTA2* expression was significantly decreased (Figure 4E). The effects of the selective CCK receptor antagonists on hPSC expression were variable, with both the CCK-AR and CCK-BR antagonists decreasing the expression of *EPHB2*, *GLI*-*3*, and *ACTA2,* whereas only the CCK-BR antagonist treatment decreased the expression of *IL*-*8* and *COLO1α1* and only the CCK-AR antagonist had modest effects on *RICTOR*.

### 3.4. Evaluation of mPSCs Protein Activation by Western Blotting

Analysis of fibrosis-associated protein activation in mPSCs through western immunoblotting was performed for collagen1α1. A diagram of mature collagen and precursor collagen protein with size is shown in Figure 5A [54]. A western blot gel with protein lysates from control cells, CCK, proglumide, combination, and the CCK-AR and CCK-BR antagonist-treated mPSCs immunoreacted with a Col1α1 antibody is shown in Figure 5B. The gel shows bands present for pro-collagen1α1 (top), mature collagen-1α1 (middle), and collagen −1α1 cleavage products (bottom) normalized to β-actin. Mean densitometry ratios normalized for β-actin show that pro-collagen1α1 protein expression was significantly decreased by treatment of mPSCs with proglumide (*p* < 0.00001), combination (*p* = 0.009) and less so with the CCK-BR antagonist (*p* < 0.02) compared to control mPSCs (Figure 5C), suggesting that the CCK-BR signaling pathway is involved in collagen1α1 synthesis from activated mPSCs. Analysis of the bands for mature collagen1α1 reveals that only treatment with proglumide significantly decreases protein expression. There was no inhibition of mature collagen1α1 protein expression by the selective CCK receptor antagonists, and proglumide at the 20 nM dose was unable to block the effect of mature collagen1α1 when administered in combination with CCK, the agonist (Figure 5D). Remodeling of the extracellular matrix (ECM) and collagen degradation is an important feature in decreasing the fibrosis of pancreatic cancer TME. Collagen1α1 cleavage products, as examined by western blot, show an increase in the degradation products in mPSCs treated with CCK, proglumide, and the combination (Figure 5E). Of interest, the CCK-BR antagonist inhibited collagen1α1 cleavage products, implying that this compound may not be as effective in degrading collagen in the ECM as proglumide or that proglumide increases collagen1α1 cleaved products by a receptor-independent mechanism. Western blotting of αSMA from stellate cell homogenates is shown in Figure 5F,G. The protein was normalized with β-tubulin since the β-actin band was too close to the size of the αSMA protein. The αSMA ratio expression by western immunoblotting was significantly decreased in proglumide-treated mPSCs (Figure 5G).

### 3.5. Proglumide Treated Cells Show Decreased Concentrations of Proline and 4-Hydroxyproline

As collagen is one of the predominant proteins found in the PDAC ECM, it has been shown that collagen can act as a source of proline and its derivative 4-hydroxyproline through degradation by proline oxidase (POX) during hypoxic or nutrient-low periods, promoting tumor survival [55,56]. Once proline is modified by POX, it has the ability to either enter the tricarboxylic acid (TCA) cycle through conversion to glutamate or enter the urea cycle through conversion to ornithine. With this notable bypass mechanism, we wanted to understand if proglumide treatment altered proline concentrations. UPLC-MS analysis found that both proline (*p* = 0.0003) and 4-hydroxyproline (*p* = 0.0001) concentrations significantly decreased upon proglumide treatment in mPSCs (Figure 6A,B), respectively. This change correlates with decreased *Col1α1* expression from qRT-PCR and western immunoblotting analysis, suggesting that proglumide therapy alters the pancreatic cancer ECM.

### 3.6. Proglumide Decreases PSC Proliferation

CCK treatment stimulated the proliferation of both the mouse (Figure 6C) and human PSCs (Figure 6D). The PSCs are considered to be activated when grown on plastic in culture plates, and it is remarkable that CCK can still further stimulate their growth. Since they are already in the activated state, proglumide monotherapy is able to reduce their activation by slowing proliferation in the MTT assay. Furthermore, proglumide is capable of blocking the CCK-stimulated proliferation, suggesting that CCK-mediated growth of PSC is a receptor-mediated phenomenon. The human PSCs grow slower than the mPSC, and in these cells, the effects of the agonist CCK, proglumide, or the combination of CCK and proglumide are most pronounced (Figure 6D) at 48 h.

## 4. Discussion

This research investigation demonstrates a novel approach to tackling the stroma in the pancreatic cancer tumor microenvironment through interruption of the CCK receptor signaling pathway with the CCK receptor antagonist, proglumide. Proglumide is unique compared to other CCK receptor antagonists in that it has an affinity for both CCK-A and CCK-B receptors. Others have previously described CCK-AR and CCK-BR expression in human PSCs. In this work, we also found that mouse PCSs express both CCK receptors. Some functional studies, such as pro-collagen production and migration, appear to be mediated primarily by the CCK-BR. Many of the genes involved in PSC activation of the ECM proteins are affected by the actions of both the CCK-AR and the CCK-BR. The PSCs, when grown on plastic tissue culture plates, become activated and assume the characteristics of myofibroblasts, including proliferation, migration, collagen production, and activation of genes that transform the pancreatic tumor microenvironment. Although our PSCs in culture were already considered activated, the addition of the agonist CCK further stimulated the cells modestly. However, since they were already activated, the application of CCK receptor antagonists served to deactivate the cells or induce plasticity, reverting them to a more quiescent state.

In the pancreatic cancer TME, it is known that various populations of fibroblasts exist, with many arising from the resident PSCs and others perhaps migrating to the pancreas. In the complex pancreatic cancer microenvironment, the activated PSCs or transformed fibroblasts (also called cancer-associated fibroblasts, CAFs) have a heterogeneous population [22], including inflammatory CAFs, myofibroblast CAFs, and even antigen-presenting CAFs [22,27]. In addition to the deposition of collagen in the pancreas extracellular matrix, these activated PSCs have several other important roles, including promoting tumor growth and metastasis and establishing an immuno-suppressive microenvironment. Most studies have demonstrated that the dense fibrosis in pancreatic cancer provides a growth advantage to the tumor and prevents the penetration of chemotherapeutic drugs and CD8+ T-lymphocytes. However, some therapeutic strategies aiming to deplete fibroblasts have paradoxically been shown to contribute to the aggressiveness of PDAC rather than to therapeutic benefits. For example, the elimination of sonic hedgehog (SHH) signaling eliminates the myofibroblasts within the TME, and this complete depletion renders the tumor more aggressive and metastatic [28]. The mechanism underlying the increased invasive nature with depletion of SHH was found to be due to the loss of the Rho effector protein kinase N2 (PKN2), which is critical for PSC myofibroblast differentiation [57].

The primary function of αSMA is not the formation of fibrosis but of mechanical tension in tissue [58]. Studies have reported that αSMA is an inconsistent marker of contractile and collagen-producing fibroblasts [59]. In addition to being a marker of myofibroblasts, αSMA is also expressed by activated PSCs and not in quiescent cells [60]. Human PSCs associated with pancreatic cancer, or CAFs, are known to be heterogeneous, with subtypes characterized by high versus low αSMA expression [61]. Quiescent hPSC do not express αSMA [62].

Novel strategies are needed to safely convert activated PSC to quiescent cells without disruption of normal physiologic pathways. There is convincing evidence that therapeutic strategies that aim to reprogram activated PSCs rather than eliminate these cells hold great promise. Schnittert and colleagues [63] suggest that approaches should focus on either inhibiting the activation of quiescent PSCs into CAFs or reverting the activated PSCs to a more quiescent phenotype. Evidence indicates that CAF subtypes are dynamic and exhibit plasticity [64] with the ability to interconvert depending on prompts from tumor cells, culture conditions, and therapeutic regimens. We have utilized several techniques to demonstrate that proglumide alters the plasticity of the PSCs, inducing these cells to a more quiescent phenotype. For example, the migration assay study shows that proglumide treatment could decrease the rate of PSC migration. The slowed migration, even in cells treated with CCK, indicates its potential role in altering cell plasticity to a more quiescent state. Notably, proglumide therapy also resulted in significant changes in Col1α1 expression. It has been noted that Col1α1 is often associated with proliferation and a worse prognosis in pancreatic cancer [49]. Treatment with both proglumide and the selective CCK-BR antagonist showed significant decreases in procollagen1α1 expression through western immunoblotting, and proglumide decreased *Col1α1* expression through qRT-PCR, indicating that collagen1α is a target of proglumide.

We found that proline and 4-hydroxyproline levels were also decreased with proglumide treatment. Proline is a major component of collagen; hence a decrease in levels would be compatible with the decrease in fibrosis we observed in pancreatic tumors treated with proglumide. Furthermore, proline also serves as a potential source of energy to fuel the tumor in the hypoxic pancreatic tumor microenvironment [55,56]; therefore, a reduction in proline availability may impair the metabolism of the tumor cells and their ability to survive in a hypoxic environment. Serum measurements of 4-hydroxyproline have been utilized in the clinic to determine tissue fibrosis and are also being used as a fibrosis biomarker for response to therapy. We have reported that 4-hydroxyproline serum levels decreased in human subjects treated with proglumide, and this decrease corresponded to a reduction in hepatic fibrosis [42].

There are many potential advantages of using proglumide for the treatment of pancreatic cancer. It has been well documented that there is cross-talk in the pancreatic tumor microenvironment and cooperation between the stellate cells and the cancer epithelial cells [7]. Many studies that have targeted only the PSCs or only the cancer epithelial cells have proven to be suboptimal in the clinic, and new therapeutic options are needed [6]. Interruption of the CCK-BR signaling pathway with proglumide not only inhibits the proliferation of pancreatic cancer epithelial cells and tumors [39] but also decreases the function of activated PSCs, thus providing a two-for-one hit therapeutic for pancreatic cancer. With this dual activity, we have previously shown that proglumide not only decreases pancreatic tumor growth but decreases tumoral fibrosis while decreasing metastases and improving survival [38,39]. Proglumide is a safe, oral medication with rapid bioavailability that has been used in the clinic [42].

Although pharmacologic studies show that proglumide is a weaker CCK receptor antagonist compared to the more selective CCK-BR antagonist [36], we clearly demonstrated the advantages of using proglumide over the highly selective CCK-BR antagonist L365,260. Treatment of PSCs with CCK showed activation of downstream signaling through AKT phosphorylation. Some increase in phospho-AKT was also observed on the western blot after PSC were treated with the CCK-BR antagonist, L365,260, yet no activation was seen with proglumide therapy. These data would imply that after receptor interaction, the highly selective antagonist could potentially stimulate intracellular signal transduction. Proglumide, in contrast, inhibits agonist binding (i.e., CCK) to the receptor without activating signal transduction. Other evidence that proglumide treatment would be more advantageous compared to the highly selective CCK-BR antagonist was that proglumide treatment had anti-fibrinolytic properties with increased expression of collagen1α1 cleavage products identified by western blot, whereas the selective CCK-BR antagonist decreased expression of the cleavage products. Highly selective CCK-AR receptor antagonists have been shown to inhibit gallbladder contraction and increase the risk of cholecystitis [65].

## 5. Conclusions

In conclusion, innovative strategies are needed to improve the survival rates of pancreatic cancer patients. The CCK receptor antagonist proglumide inhibits pancreatic tumoral fibrosis by its interactions with PSC, rendering them more quiescent without eliminating their innate function. These properties, along with the anti-cancer epithelial growth effects of proglumide, explain the mechanisms by which this CCK receptor antagonist decreases pancreatic cancer growth and metastases and prolongs survival [39]. Demonstrating the anti-cancer properties in both mouse and human PSC and in our current and prior work in human pancreatic tumors suggests proglumide may be a safe and effective treatment option for human subjects with pancreatic cancer.

## 6. Patents

Georgetown University holds a patent titled ‘Treating Cancer with a CCK Receptor Inhibitor and an Immune Checkpoint Inhibitor’. US Patent Application #16/493,882 was filed on 13 September 2019, and issued as a US patent No. 11,278,551 on 22 March 2022.

## Figures and Tables

**Figure 1 cancers-15-02811-f001:**
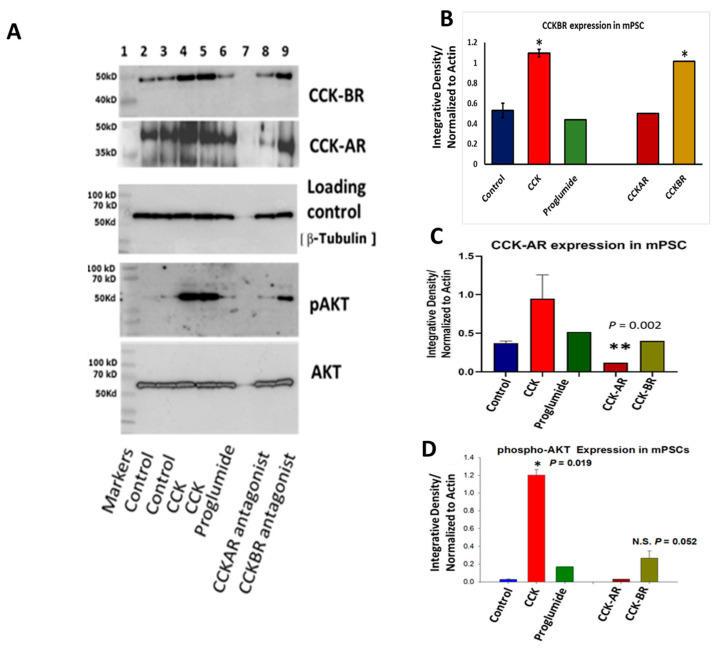
CCK receptor expression in PSCs. (**A**) Western blot of mPSC protein lysates after treatment with CCK peptide or receptor antagonists reveals CCK-BR and CCK-AR expression and activation of downstream signaling phospho-AKT. (**B**) Densitometry analysis ratios normalized to β-actin of CCK-BR expression show that treatment with CCK and the CCK-BR antagonist L365,260 upregulate the protein expression of the CCK-BR. (**C**) Densitometry analysis ratios normalized to actin of CCK-AR expression show decreased CCK-AR expression in cells incubated with the CCK-AR antagonist. (**D**) Densitometry ratios of phosphorylated AKT after stimulation or receptor antagonism normalized to total AKT show increased phosphorylation of AKT in CCK-treated mPSC. Note CCK-AR and CCK-BR refer to antagonist-treated cells. N.S. = not significant; significantly different from control * *p* < 0.05; ** *p* < 0.01. The original western blot figures could be found in Appendix A.

**Figure 2 cancers-15-02811-f002:**
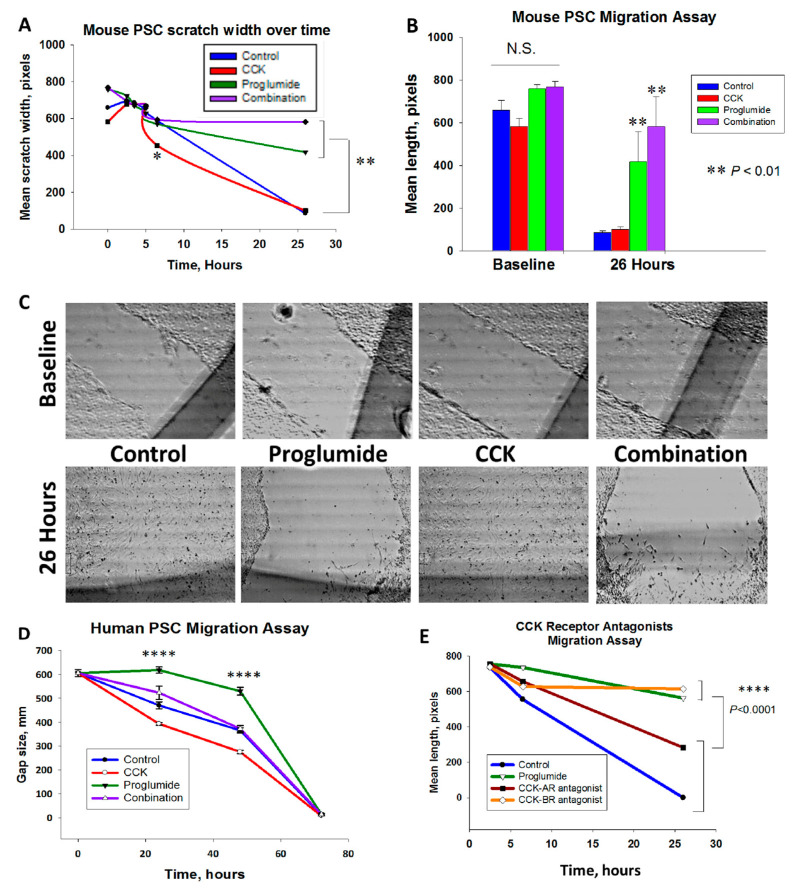
Migration of pancreatic stellate cells is mediated through the CCK-BR. (**A**) Mean scratch width of mPSC over 26 h shows an initial increase in migration rate in CCK-treated cells and inhibition of the migration by proglumide. Significantly different from controls * *p* = 0.01; ** *p* < 0.01. (**B**) Baseline and 26 h mean scratch width ± SEM for each treatment group. (**C**) Representative photos from each treatment group show the baseline scratch (top) and after 26 h of migration (bottom). (**D**) Human PSC migration assay with mean scratch width ± SEM for each treatment group at four time points; baseline, 24, 48, and 72 h. (**E**) Time course of mPSC migration assay in untreated cells and mPSCs treated with proglumide, the CCK-AR antagonist, or the CCK-BR antagonist. N.S. = not significant. Significantly different from controls * *p* < 0.05; ** *p* < 0.01; **** *p* < 0.0001.

**Figure 3 cancers-15-02811-f003:**
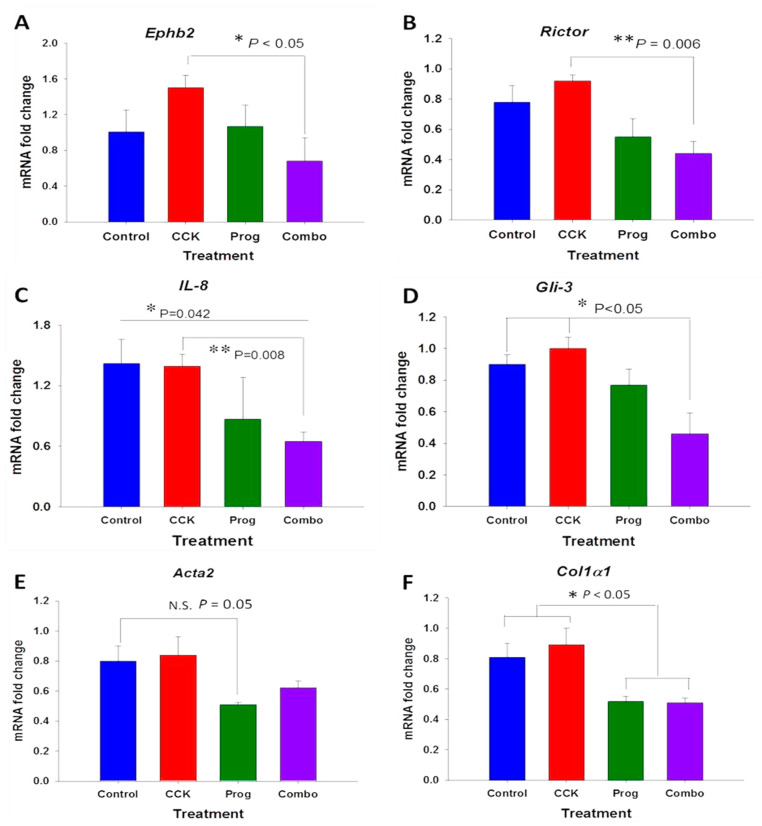
Differentially expressed genes in mPSC treated with proglumide, CCK, or the combination of CCK and proglumide. (**A**) Proglumide decreases the expression of *Ephb2* in CCK-stimulated mPSC. Ephrin type-B receptor 2 is a gene that codes for a tyrosine kinase receptor implicated in tissue fibrosis. (**B**) *Rictor*/mTORC2 signaling mediates TGFβ1-induced fibroblast activation. Proglumide therapy downregulates CCK-activated *Rictor* expression. (**C**) The inflammatory cytokine *IL*-*8* mRNA expression is significantly downregulated by the combination of CCK and proglumide. (**D**) Proglumide significantly downregulates the expression of CCK-stimulated *Gli3*, a transcriptional regulator of the Hedgehog (Hh) signaling pathway involved in tissue fibrosis. (**E**) Proglumide monotherapy does not significantly downregulate *Acta2* expression, the gene responsible for alpha-smooth muscle actin (αSMA). (**F**) Proglumide decreases the expression of CCK-stimulated and unstimulated *Col1α1* (which codes for collagen type 1α chain 1), a major component of the pancreatic cancer tumor microenvironment. Prog = proglumide; Combo = combination therapy with proglumide and CCK. N.S. = not significant. Significant different, * *p* ≤ 0.05; ** *p* < 0.01.

**Figure 4 cancers-15-02811-f004:**
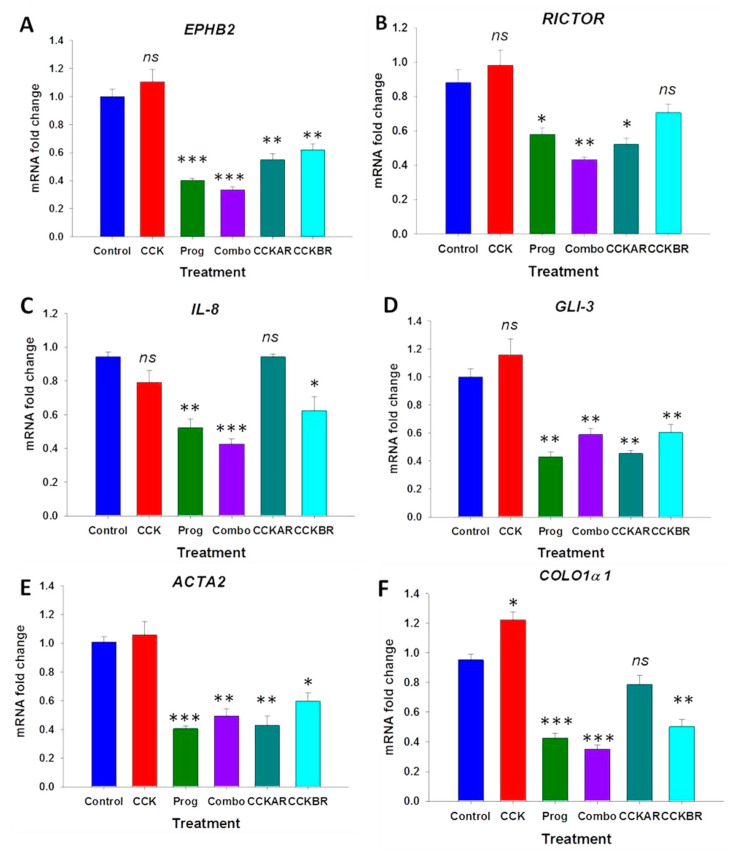
Differentially expressed genes in human PSC treated with media alone (control), CCK, proglumide, the combination of CCK and proglumide, the selective CCK-AR antagonist, and the CCK-BR antagonist. (**A**) Proglumide decreases expression of *EPHB2* expression alone and in combination with CCK. CCK-AR and CCK-BR antagonist treatment decrease *EPHB2* expression. (**B**) *RICTOR* expression is downregulated in hPSCs with proglumide alone or co-treated with CCK. *RICTOR* expression is also decreased by the CCK-AR antagonist. (**C**) *IL*-*8* mRNA expression is significantly downregulated in hPSCs treated with proglumide and the CCK-BR antagonist. (**D**) Proglumide and both selective CCK receptor antagonists significantly downregulate the expression of *GLI3*. (**E**) Treatment of hPSCs with proglumide and the selective CCK receptor antagonists downregulate *ACTA2* expression. (**F**) Proglumide and the CCK-BR antagonist decrease mRNA expression of *COL1α1*. Prog = proglumide; Combo = combination therapy with proglumide and CCK. *ns* = not significant. Significant different, * *p* < 0.05; ** *p* < 0.01; and *** *p* < 0.005.

**Figure 5 cancers-15-02811-f005:**
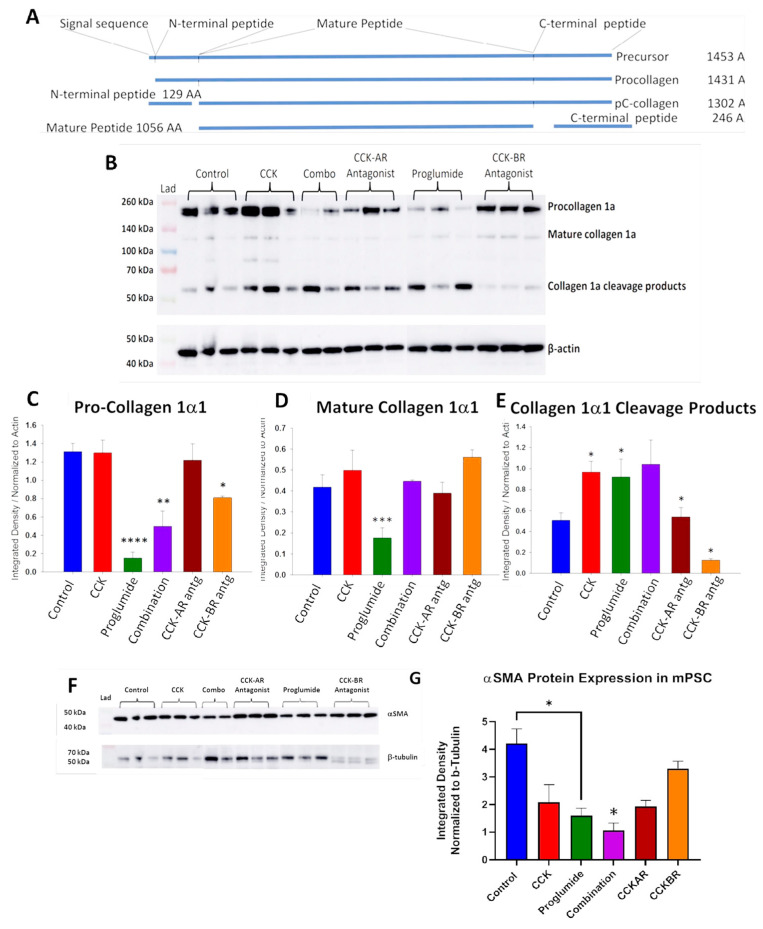
Fibrosis-associated proteins in CCK-BR signaling pathway in mPSCs by western immunoblotting. (**A**) Diagram of mature collagen and precursor collagen protein with size (Reprinted and adapted with permission from Iannarone et al. [54]). (**B**) Western blot of protein lysates of mPSCs treated with CCK, proglumide, the combination of CCK and proglumide, the selective CCK-AR antagonist (L364,718), and the selective CCK-BR antagonist (L365,260) probed with a polyclonal antibody for Col1α1 and normalized with β-Actin. The gel shows bands present for pro-collagen1α1 (top), mature collagen1α1 (middle), and collagen-1α1 cleavage products (bottom). Note this gel was run on two separate apparatuses due to the large number of wells required, and the gels are fused for appearance. (**C**) Densitometry normalized to β-Actin of the Pro-collagen1α1 ratio expression was significantly decreased in the proglumide (*p* < 0.00001), combination (*p* = 0.009), and CCK-BR antagonist (*p* < 0.02) compared to control. (**D**) Densitometry of the gel bands for mature collagen shows that protein expression was significantly decreased by treatment of the mPSCs with proglumide (*p* < 0.005). (**E**) Collagen-1α1 cleavage products were significantly increased by CCK, proglumide, and the combination of CCK and proglumide. (**F**) Western blot of mPSC protein lysates reacted with αSMA antibody and normalized to β-tubulin. Note this gel was run on two separate apparatuses due to the large number of wells required, and the gels are fused for appearance. (**G**) Columns represent means ± SEM of densitometry ratios (*n* = 3). Significant change in αSMA protein expression was seen in proglumide-treated mPSCs. Significantly different from controls * *p* < 0.05; ** *p* < 0.01; *** *p* < 0.005; and **** *p* < 0.0001. The original western blot figures could be found in Appendix A.

**Figure 6 cancers-15-02811-f006:**
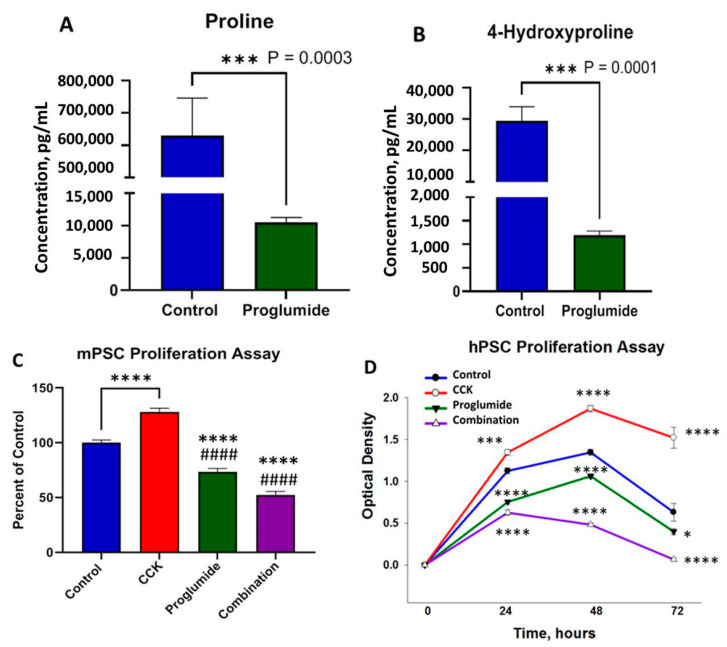
Effects of proglumide treatment on proline and 4-hydroxyproline levels, and proliferation of pancreatic stellate cells. (**A**) Concentrations of proline measured by mass spectroscopy were significantly lower in proglumide-treated mPSCs *** *p* = 0.0003. (**B**) Concentrations of 4-hydroxyproline were significantly decreased in proglumide-treated mPSCs *** *p* = 0.0001. (**C**) Cell proliferation of mPSCs was increased by CCK (10 nM) treatment after 24 h of exposure. Proglumide (20 nM) monotherapy or in combination with CCK decreased proliferation. (**D**) Cell proliferation of hPSCs over time treated with CCK (10 nM), proglumide (20 nM), or the combination of CCK and proglumide compared to control (untreated cells). Significantly different from control * *p* < 0.05, ***, *p* < 0.005, **** *p* < 0.0001, and significantly different from CCK-treated cells ^####^, *p* < 0.0001.

**Table 1 cancers-15-02811-t001:** Primer sequences for mouse pancreatic extracellular matrix-associated genes.

*mGene*	Forward 5′-3′	Reverse 5′-3′
*mEphb2*	CAACGGTGTGATCCTGGACTAC	CACCTGGAAGACATAGATGGCG
*mHic-5*	GGTCTGGAGAATCTTCAGGAACC	CACCACTGGAAGAGGAGAATGG
*mGli3*	CTGCGGTATCTCCTCTCATAGG	CAGCACTGTGAAGTCTACACCTG
*mRheb*	GGCAAGTTGTTGGATATGGTGGG	CCAAGATTCTGCCAAAGCCTTTC
*mRictor*	CAGTGTGAGGTCCTTTCCATCC	GCCATAGATGCTTGCGACTGTG
*mFap*	CACCTGATCGGCAATTTGTG	CCCATTCTGAAGGTCGTAGATGT
*mIL-1β*	GGACCTTCCAGGATGAGGACA	GTTCATCTCGGAGCCTGTAGTG
*mIL-8*	GGTGATATTCGAGACCATTTACTG	GCCAACAGTAGCCTTCACCCAT
*mCol1a1*	CGCCATCAAGGTCTACTG	ACGGGAATCCATCGGTC
*mCol4a*	GATGGGCTATCCTGGAACCACT	TCTCTCCTCGTTCGCCTTTGG
*mActa2*	TGCCGAGCGTGAGATTGT	CCCGTCAGGCAGTTCGTAG

**Table 2 cancers-15-02811-t002:** Primer sequences for human pancreatic extracellular matrix-associated genes.

	Forward (5′-3′)	Reverse (5′-3′)
*hRICTOR*	GCCAAACAGCTCACGGTTGTAG	CCAGATGAAGCATTGAGCCACTG
*hEPHB2*	CGCCATCTATGTCTTCCAGGTG	GATGAGTGGCAACTTCTCCTGG
*hIL-8*	GAGAGTGATTGAGAGTGGACCAC	CACAACCCTCTGCACCCAGTTT
*hGLI3*	TCAGCAAGTGGCTCCTATGGTC	GCTCTGTTGTCGGCTTAGGATC
*hACTA2*	CTATGCCTCTGGACGCACAACT	CAGATCCAGACGCATGATGGCA
*hCOL1A1*	GATTCCCTGGACCTAAAGGTGC	AGCCTCTCCATCTTTGCCAGCA

## Data Availability

Data is available in the Appendix A or from the corresponding author upon request.

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
