# Peer review of "Cholecystokinin Receptor Antagonist Induces Pancreatic Stellate Cell Plasticity Rendering the Tumor Microenvironment Less Oncogenic"

_cancers, 2023, doi:10.3390/cancers15102811_

Round 1

Reviewer 1 Report

The manuscript by Jolly and Dunka aims to demonstrate the role of cholecystokinin receptor antagonists in reducing the activation of pancreatic stellate cells, specifically by inhibiting the CCK-RB receptor. While this line of research could be interesting for the scientific community, the experiments presented by the authors are often unclear and contain significant discrepancies that make it difficult to trust the results obtained. Moreover, key experiments demonstrating the selective role of CCK-RB over CCK-RA are missing.

Major:

  1. Data displayed in Figure 2 are inconsistent between murine and human PSC. For murine PSC, the combination of CCK (pro-migration) and Proglumide (anti-migration) results in greater migration inhibition than Proglumide alone. Instead, in human PSC, the combination treatment with CCK and Proglumide did not result in any appreciable difference over CCK treatment alone. This significant inconsistency makes it difficult to interpret or fully trust the manuscript's conclusion.
  2. To validate the key role of CCK-RB over CCK-RA, experiments described in Figures 3 and 4 must be replicated using selective inhibitors. Additionally, the combo treatment must be performed with selective inhibitors.

Minor:

  1. Graphs in Figures 1A and 1B are difficult to interpret. Which cells are used as controls? In line 272-274, the authors make a comparison with "harvested normal mouse pancreas" and "total human pancreas" in line 279. If this is true, it is not an appropriate control since comparing lysate from a tissue lysate and a cultured cell line is not meaningful. To confirm the expression of CCK-A and CCK-B in murine pancreatic stellate cells, the authors need to rely on IHC or in situ hybridization.
  2. The authors tend to be extremely repetitive, which does not improve the interpretation of the results. For example, from line 305 to 321, the authors describe the results of a single experiment three times. Figures 2A, 2B, and 2C are just different graphical representations of the same experiment and must be described once.
  3. Professional English correction services are strongly recommended to improve readability throughout the manuscript. For example, the sentence in line 351-352 makes no sense.

Author Response

Reviewer 1:

Comments and Suggestions for Authors

The manuscript by Jolly and Duka aims to demonstrate the role of cholecystokinin receptor antagonists in reducing the activation of pancreatic stellate cells, specifically by inhibiting the CCK-RB receptor. While this line of research could be interesting for the scientific community, the experiments presented by the authors are often unclear and contain significant discrepancies that make it difficult to trust the results obtained. Moreover, key experiments demonstrating the selective role of CCK-RB over CCK-RA are missing.

Major:

  1. Data displayed in Figure 2 are inconsistent between murine and human PSC. For murine PSC, the combination of CCK (pro-migration) and Proglumide (anti-migration) results in greater migration inhibition than Proglumide alone. Instead, in human PSC, the combination treatment with CCK and Proglumide did not result in any appreciable difference over CCK treatment alone. This significant inconsistency makes it difficult to interpret or fully trust the manuscript's conclusion.

Response: The original figures were reanalyzed and statistics done by GraphPad Prism. We have change the figures to a line graph over time to better represent the changes seen. Also representative images from each treatment group are now provided in the Supplemental data.

2. To validate the key role of CCK-BR over CCK-AR, experiments described in Figures 3 and 4 must be replicated using selective inhibitors. Additionally, the combo treatment must be performed with selective inhibitors.

Response: The PSC are already considered to be activated when they are grown on tissue culture plates. . Since the cells are already activated when grown on plastic culture plates, we only observed a modest stimulatory effect with the addition of CCK as well as activation of phosphor-AKT. These studies support that proglumide was acting through a CCK receptor mediated pathway. But the purpose of this work was not to try to further activate PSCs but to de-activate them  and to test the CCK receptor antagonist proglumide as a novel agent to induce PSC plasticity. At the reviewer’s request, we have now repeated all the PCR experiments in hPSCs with proglumide and the CCKAR and CCKBR antagonists. Figure 4 is a new figure. It is already known that proglumide has affinity for both receptors. In this work we show that proglumide reverts PSC to a quiescent phenotype. Some of the actions are mediated by the CCKAR and some by the CCKBR and some by both. We include in the discussion rationale for using proglumide rather than the highly selective antagonists.

Minor:

  1. Graphs in Figures 1A and 1B are difficult to interpret. Which cells are used as controls? In line 272-274, the authors make a comparison with "harvested normal mouse pancreas" and "total human pancreas" in line 279. If this is true, it is not an appropriate control since comparing lysate from a tissue lysate and a cultured cell line is not meaningful. To confirm the expression of CCK-A and CCK-B in murine pancreatic stellate cells, the authors need to rely on IHC or in situ hybridization.

Response: We agree with the reviewers in that using whole mouse or human pancreas homogenates as a control is not entirely accurate and may obscure the data from isolate pancreatic stellate cells. Others have previously published data showing CCK-AR and CCK-AR expression in rat and in human PSC (see references in manuscript by Berna and Phillips); therefore we removed these PCR figures about receptor expression. Receptor protein expression was confirmed by using the mouse PSC homogenates by western blot in revised Figure 1A. We have now added a western blot with CCK-AR expression to this figure. This is the first publication we are aware of showing CCK receptor protein expression in murine PSC. We did not repeat work done by others (references by Berna and Phillips) using human or rat PSCs.

2. The authors tend to be extremely repetitive, which does not improve the interpretation of the results. For example, from line 305 to 321, the authors describe the results of a single experiment three times. Figures 2A, 2B, and 2C are just different graphical representations of the same experiment and must be described once.

Response: This section has been shortened and is no longer repetitive.

3. Professional English correction services are strongly recommended to improve readability throughout the manuscript. For example, the sentence in line 351-352 makes no sense.

Response: All the spelling and grammar were double checked and accurate.

Reviewer 2 Report

In this article, the authors studied the effects of the CCK peptide agonist and CCK receptor antagonists on activation, migration, collagen deposition, and proliferation in both murine and human PSCs (pancreatic stellate cells). The results suggest that a CCK receptor antagonist, proglumide, targeting the CCK-B receptor signaling pathway may alter plasticity of PSC rendering them more quiescent and leading to a decrease in fibrosis in the pancreatic cancer microenvironment. Proglumide may be a safe and effective treatment for pancreatic cancer patients. Although the authors emphasized this work has implication for the treatment of pancreatic cancer patients, there are a number of critical problems, as described below.

1.       In Results 3.1, protein expression of CCK-AR in mPSCs was not shown. Protein expression of CCK-AR, CCK-BR or downstream AKT after various treatments in hPSCs must be investigated. In addition, in Figure 1C, the bands of AKT were supersaturated and overexposed, a more appropriate picture is needed.

2.       In Results 3.2, in addition to wound-healing assay, transwell assay must be used to evaluate cell migration ability. In Figure 2D-2Erepresentative images from the migration assay must be shown, and better images are needed.

3.       In Results 3.3 and Results 3.4, expression of specific differently genes involved in fibrosis and fibrosis-associated protein activation after various treatments in mPSCs were investigated, yet how about in hPSCs?

4.       In Results 3.6, to better illustrate cell growth rates, growth curves at different time points after different treatments are need, e.g., 24h, 48h, 72h, 96h.

Author Response

Reviewer 2

Comments and Suggestions for Authors

In this article, the authors studied the effects of the CCK peptide agonist and CCK receptor antagonists on activation, migration, collagen deposition, and proliferation in both murine and human PSCs (pancreatic stellate cells). The results suggest that a CCK receptor antagonist, proglumide, targeting the CCK-B receptor signaling pathway may alter plasticity of PSC rendering them more quiescent and leading to a decrease in fibrosis in the pancreatic cancer microenvironment. Proglumide may be a safe and effective treatment for pancreatic cancer patients. Although the authors emphasized this work has implication for the treatment of pancreatic cancer patients, there are a number of critical problems, as described below.

  1. In Results 3.1, protein expression of CCK-AR in mPSCs was not shown. Protein expression of CCK-AR, CCK-BR or downstream AKT after various treatments in hPSCs must be investigated. In addition, in Figure 1C, the bands of AKT were supersaturated and overexposed, a more appropriate picture is needed.

Response: Figure 1 is revised and the western blot for the CCK-AR expression is added to the revised Fig 1A. A revised figure for AKT was applied.

2. In Results 3.2, in addition to wound-healing assay, transwell assay must be used to evaluate cell migration ability.

Response: The scratch test is a standard protocol that is published in hundreds of peer reviewed manuscripts. Transwells are often used for co-culture experiments.

3. In Figure 2D-2E,representative images from the migration assay must be shown, and better images are needed.

Response: Representative images for each treatment over time is now included for the HPSCs in Supplemental Figure S4 and representative photos for all the treatments over time are now included for the antagonist migration assay and included in the revised manuscript in Supplemental Figure S5. Figures were converted to High resolution (300dpi) TIFF or jpeg files

4. In Results 3.3 and Results 3.4, expression of specific differently genes involved in fibrosis and fibrosis-associated protein activation after various treatments in mPSCs were investigated, yet how about in hPSCs?

Response: We have now reordered human primers and repeated the entire PCR section for differentially expressed genes in the hPSC – this is revised Figure 4.

5. In Results 3.6, to better illustrate cell growth rates, growth curves at different time points after different treatments are need, e.g., 24h, 48h, 72h, 96h.

Response: Growth rates in hPSCs are now plotted over time up to 72 hrs and the graph was changed to a line graph growth rates rather than columns as recommended by the reviewer. This is now the revised figure 6.D